# Environmental DNA: A promising factor for tuberculosis risk assessment in multi-host settings

Jordi Martínez-Guijosa[1]*, Beatriz Romero[2], José Antonio Infantes-Lorenzo[2,3], Elena Díez[4], Mariana Boadella[5], Ana Balseiro[6], Miguel Veiga[1], David Navarro[4], Inmaculada Moreno[7], Javier Ferreres[1], Mercedes Domínguez[7], Cesar Fernández[4], Lucas Domínguez[2,3], Christian Gortázar[1]

**1** SaBio, IREC (UCLM-CSIC-JCCM), Ciudad Real, Spain, **2** VISAVET Health Surveillance Centre, Complutense University of Madrid, Madrid, Spain, **3** Faculty of Veterinary Medicine, Department of Animal Health, Complutense University of Madrid, Madrid, Spain, **4** Livestock Service, Department of Rural Development, Environment and Local Administration, Government of Navarra, Pamplona, Navarra, Spain, **5** SABIOTEC, Edificio Polivalente UCLM, Ciudad Real, Spain, **6** SERIDA, Regional Service for Agrifood Research and Development, Asturias, Spain, **7** Area of Immunology, Microbial and Immunogenetic Immunology Unit, Instituto de Salud Carlos III, Majadahonda, Madrid, Spain

☯ These authors contributed equally to this work.
* jordi.m.guijosa@gmail.com

**Data Availability Statement:** The datasets generated for this study can be found in the Mendeley Data repository (Martinez-Guijosa, Jordi (2019), "DB Environmental DNA Navarra",

## Abstract

Attaining and maintaining the Official Tuberculosis Free status continues to be a challenge when several domestic and wild hosts contribute to the maintenance of the *Mycobacterium tuberculosis* complex (MTC). Local tuberculosis hotspots are sometimes identified in cattle in low-prevalence regions. We have, therefore, studied one such hotspot in depth in order to produce an epidemiological diagnosis. Host population size and MTC prevalence were estimated in selected wildlife and in livestock, while on-cattle environmental DNA detection was additionally used as a proxy for risk of exposure at the farm (herd) level. Positive skin test reactors were found on 16 of the 24 cattle farms studied in the period 2012–2016. Although all goats tested negative to the skin test during this period, MTC was confirmed in four sheep at slaughter, thus indicating an unknown prevalence of infection in this host species. With regard to wildlife, the prevalence of MTC infection based on culture was 8.8% in the case of wild boar (*Sus scrofa*), and the only road-killed badger (*Meles meles*) submitted for culture tested positive. Two criteria were employed to divide the cattle farms into higher or lower risk: tuberculosis testing results and environmental DNA detection. Environmental MTC DNA detection yielded significant differences regarding "use of regional pastures" and "proximity to woodland". This study suggests that on-animal environmental DNA sampling may help when assessing contact risk as regards MTC in livestock at the herd level. This tool opens up new avenues of epidemiological research in complex multi-host settings.

Mendeley Data, v1. http://dx.doi.org/10.17632/2xhxw55wny.1). Molecular epidemiology data is also available in mycoDB.es.

**Funding:** This work was supported by the Programa de Tecnologías Avanzadas en Vigilancia Sanitaria (TAVS) from the Comunidad de Madrid (ref. S2013/ABI-2747). JMG holds a FPI pre-doctoral scholarship (BES-2015- 072206), funded by MINECO. This is a contribution to MINECO grant CGL2017-89866 WildDriver and EU-FEDER. This is also a contribution to Valle de Alcudia pilot project SG-2019-02 from PDR-CLM, and to GOSTU project EU-FEADER (AEIAGRI-PNDR-MAPA, ref. 20190020007521). The funders had no role in study design, data collection and analysis, decision to publish, or preparation of the manuscript.

**Competing interests:** I have read the journal's policy and the authors of this manuscript have the following competing interests: MB and SABIOTEC are consultants and offer risk assessment at the wildlife-livestock interface. This does not alter our adherence to PLOS ONE policies on sharing data and materials.

## Introduction

Animal tuberculosis (TB) is a multi-host infection caused by *Mycobacterium bovis* and closely related members of the *Mycobacterium tuberculosis* complex (MTC). Several EU-member states, such as France, Germany or Poland, are officially TB-free (OTF), signifying that less than 0.1% of their herds of cattle are still infected. For a Member State or region to achieve OTF status as defined in Council Directive 64/432/EEC, at least 99.9% of the herds of cattle within it must have been or remained OTF for at least six consecutive years. Eradication would represent the total elimination of the infection. However, achieving eradication or attaining and maintaining the OTF status continues to be a challenge when several domestic and wild hosts contribute to MTC maintenance [1].

There is still a high prevalence of TB in the south-western quarter of the Iberian Peninsula, where over 5% of herds of cattle are infected [2], the infection is present in other livestock [3–5], and there is an extremely high prevalence in wildlife, e.g. over 50% in Eurasian wild boar (*Sus scrofa*) and over 10% in red deer (*Cervus elaphus*) [6]. In other regions, however, there is a very low prevalence of TB in cattle, and it is limited in wildlife [7]. Moreover, the island regions are almost TB-free, partly owing to the absence of wildlife reservoirs [8].

There is, in fact, a low prevalence of TB in the cattle in most of the northern regions of the Spanish mainland, which are characterized by an Atlantic climate (herd prevalence of between 0.05 and 0.73% in Galicia and Aragón, respectively; year 2016 [2]). No TB has been detected in red deer in the same regions, despite annual sampling, and TB prevalence remains relatively low in wild boar (<10%)[7,9]. European badgers (*Meles meles*) are also suitable MTC maintenance hosts, not only in the British Isles [10], but also in parts of continental Europe [11]. With regard to the Iberian Peninsula, badgers are more abundant in Atlantic than in Mediterranean regions [12], where their prevalence is reported to be around 12% [13]. Both badgers and wild boar could, therefore, contribute to MTC maintenance in Atlantic habitats in northern Spain. Roe deer (*Capreolus capreolus*) is an abundant cervid in northern Spain [14], but despite the fact that cases of infected roe deer have been evidenced, it does not appear to be a significant TB reservoir [15,16]. Although less common in the Atlantic region, Iberian ibex (*Capra pyrenaica*) does not appear to play a role in TB maintenance in the wild either [17]. Domestic sheep [5] and goats [7] are, however, recognized TB reservoirs in the Atlantic region.

Local TB hotspots, i.e., areas of elevated disease incidence or prevalence, or a geographic cluster of cases [18], are sometimes identified in these low-prevalence Atlantic settings. We have, therefore, studied one of these hotspots in depth in order to produce an epidemiological diagnosis (ED). This ED is a step proposed by the Spanish Wildlife TB Control Plan (PATUBES [19]) prior to any intervention. An ED consists of identifying the species involved and their TB status, along with their distribution, abundance and management within the study area. In this specific case, we have generated information on wildlife and livestock numbers and their MTC infection status, carried out interviews with farmers and used participatory Geographic Information System (GIS) [20] to investigate farm-related risk factors for the incidence of TB in cattle. The innovative nature of this study lies in the use of environmental DNA detection as a proxy for the risk of exposure at the farm (herd) level. Environmental DNA detection is a suitable tool by which to detect contamination with MTC at waterholes and in other substrates [21–23]. We hypothesized that the detection of environmental DNA in herds of cattle would contribute to risk factor identification at the herd level.

## Materials and methods

### Study area

The study area is located in the north of the Iberian Peninsula, between Atlantic Spain and the pre-Pyrenean north plateau (centroid coordinates, utm wgs84: 592425, 4743194), to the northwest of Pamplona, in the region of Navarra. This area is a pre-Pyrenean basin that has a marked altitudinal gradient and several rivers and tributaries that form abundant ravines, and is characterized by the presence of the Sierra Urbasa-Andía (a mountainous Natural Park) to the west. It occupies an area of approximately 100 $Km^2$, with an average altitude of 830 meters (range: 384-1260m) and an average annual temperature of 8° - 12°C. The livestock management system employed in the area is largely determined by a marked seasonal variation in temperatures and rainfall. With regard to the monthly precipitations, the maximum are collected in December and the minimum in July. The dominant vegetation comprises Portuguese oak (*Quercus faginea*) and holm oak (*Quercus rotundifolia*) forests, and those of beech (*Fagus sylvatica*) in higher or shady areas. The forests occupy 45% of the area, and cover two thirds of the total surface if scrublands are considered. The remaining third is occupied by pastures, meadows and arable crops.

The land use in the study area is principally determined by the type of property, of which there are three different categories: private land, municipal land and regional land [24]. The private land, which is owned by farmers, is where the barns and warehouses are located. Each municipality has communal land, to which every inhabitant of each respective village has access in order to graze and water their livestock. The regional land (Urbasa-Andía Natural Park) has similar rights to those of the municipal land, with the difference that their use extends to all the farmers in the region.

### Wildlife densities and TB status

Wild boar densities were assessed using two means. On the one hand, we used three 4x4 photo-trapping grids (16 cameras each, with a distance of 1km between cameras; S1 Fig), installed from September 5 to October 12 (2016), to estimate densities on the basis of an algorithm described previously [25], based on the likelihood of randomly detecting unmarked individuals. We assumed a mean daily speed of 0.115 m/s (own data), a detection radius of 10m and a detection angle of 35°. On the other hand, we used the 2015/16 hunting data to assess the number of wild boar observed per square km. Badger densities were estimated using the same camera-trapping grids and camera specifications and a mean daily speed of 0.021 m/s. This speed was based on data obtained from radio-tracked badgers in Asturias, Spain. Fifty seven hunter-harvested wild boar and one road-killed badger were inspected for visible TB-compatible lesions [26] and submitted for bacteriological culture and molecular identification in a BSL3 laboratory, as described previously [27]. Briefly, approximately 2g of tissue sample was homogenized and decontaminated with an equal volume of 0.75% (w/v) hexadecylpyridinium chloride solution in agitation for 30 minutes. After centrifugation, pellets were collected with swabs and cultured in Löwenstein-Jensen with sodium pyruvate and Coletsos media (Difco, Spain) at 37°C for up to 3 months. A culture was considered positive when isolates were identified as MTBC by means of conventional PCR [28] and /or DVR-spoligotyping [29].

### Livestock herds, pasture management and TB status

The study area includes 36 herds of cattle with a total of 3,100 cattle, 3 herds of goats (a total of 309 goats) and 17 flocks of sheep totaling 4,786 sheep. No pigs are present. With regard to the herds of cattle, 12 were not studied because they were feedlots, because of technical restrictions

or owing to their very small size (<10 cattle). The 24 herds selected for the study comprised 2,786 cattle (89.87% of the total number of cattle in the study area). All cattle and goats are tested for TB on an annual basis by means of the official single intradermal tuberculin test and according to the Spanish Bovine Tuberculosis Eradication Program [2], and data for the period 2012–2016 was, therefore, kindly provided by the Government of Navarra.

The grazing system is complex, since herds belonging to different owners share pastures and water points. The farms that use the regional pastures (14 of the 24 farms sampled) take their livestock to this land in spring (May) in order to take advantage of these pastures before moving to municipal and private land in summer (August). Those farmers who do not use the regional pastures use the municipal pastures in their place between May and November (9 of the 24 farms sampled). In winter, when it begins to snow, the cattle are moved to stables and fed with fodder or on pastures close to the farm building. Only one farm is limited to using exclusively private land and does not share pastures with other farmers at any time.

All the work performed on the farms was carried out between September 2016 and January 2017, following the recommendations and protocols described in the Spanish Bovine Tuberculosis Eradication Program [2]. Moreover, the University of Castilla-La Mancha (UCLM) research ethics committee granted a formal waiver of ethics approval, since only routine veterinary care was involved in this study.

## On-farm risk assessment

Only two categorical variables (yes/no) were studied, namely the use of regional pastures and the use of municipal pastures. Table 1 lists the quantitative variables considered per cattle farm, and the source of the information.

## Environmental DNA

Samples of environmental DNA were taken from the cattle from 24 farms during January 2017. A dry sponge (3M™ Dry-Sponge; 3M-España, Madrid) was pre-hydrated with 15 ml of an isotonic surfactant liquid, although 15 ml of sterile distilled water could have been used as an alternative. On each farm, two sponges were scrubbed ten times on the scapula regions of two randomly chosen cattle (one sponge per animal). The scapula region, and no other anatomical region of the animal, was chosen because it is the most accessible area during cattle handling since it is relatively close to the brisket, one of the preferred areas as regards sampling pathogens on cattle hides [30]. It is additionally a large quasi-horizontal area, where dust particles may easily be deposited. Once in the laboratory, the sponge was hydrated with 10 ml of the same liquid and centrifuged at 3,500 $g$ for 10 minutes. All the steps until the recovery of DNA were performed under safe conditions in a BSL3 area.

The pellet was purified as described previously, including mechanical disruption with beads using the fast-prep equipment (before and after the enzymatic lysis) [27]. A real time PCR with which to detect $IS6110$ was used to detect MTC in the samples, using the primers described previously [31] (forward 5'- GGTAGCAGACCTCACCTATGTGT-3'; reverse 5'-AGGCGTCGGTGACAAAGG-3') and probe (5' FAM-CACGTAGGCGAACCC-MGB-NFQ 3'). The PCR was performed using the QuantiFast Pathogen +IC Kit, according to the manufacturer's instructions (Qiagen, Hilden, Germany), with a final concentration of 2.5 μM for each primer, 1.25 μM for the probe, and 5μl of the extracted DNA. All PCR reactions were carried out in a CFX96 TouchTM Real-Time PCR Detection System (Bio-Rad, Hercules, CA, USA) according to the following cycling conditions: 95˚C for 5 minutes, followed by 45 2-step cycles at 95˚C for 15 seconds and 60˚C for 30 seconds. Positive (*M. bovis* strain), and negative PCR controls (distilled water) were also included. A conventional PCR was also used in order

**Table 1. Quantitative variables analyzed, origin of the data, Z statistic and P value (Mann-Whitney's U test) for each of the two criteria defined in this study: Outbreak frequency based on TB testing results and environmental MTC DNA positivity.**

| Variable | Data source | TB testing (>1 year positive) | | MTC DNA (detected) | |
|---|---|---|---|---|---|
| | | Z | P | Z | P |
| Total N of cattle | Official data | 1.92 | 0.056 | -1.79 | 0.078 |
| Number of heifers | Official data | 1.80 | 0.076 | -1.30 | 0.19 |
| N of cattle groups per pasture | Farmer interview | 0.64 | 0.55 | 0.40 | 0.69 |
| N of imported cattle (5 years) | Official data | 0.02 | 0.98 | -0.73 | 0.47 |
| N of import movements (5 years) | Official data | 0.53 | 0.59 | 0.35 | 0.75 |
| Total land used | Farmer interview | -1.08 | 0.31 | -1.12 | 0.27 |
| Total own land | Farmer interview | 0.82 | 0.41 | 0.39 | 0.72 |
| Total own pastures | Farmer interview | -0.36 | 0.72 | 0.71 | 0.50 |
| Total own woodland | Farmer interview | -0.27 | 0.82 | -0.92 | 0.42 |
| N own goats | Farmer interview | 1.64 | 0.44 | 0.24 | 0.89 |
| N own sheep | Farmer interview | 0.65 | 0.73 | -0.51 | 0.78 |
| N own horses | Farmer interview | 1.28 | 0.20 | 0.53 | 0.65 |
| N users' municipal pastures | Farmer interview | -1.23 | 0.23 | -1.79 | 0.088 |
| **Farm altitude a.s.l.** | GIS | 0.96 | 0.35 | 2.42 | **0.014** |
| Farm distance to village | GIS | -1.35 | 0.18 | -1.68 | 0.10 |
| **Farm distance to woodland** | GIS | 1.35 | 0.18 | 2.68 | **0.006** |
| N of identified risk points | Farm visit | -0.26 | 0.81 | -1.27 | 0.21 |
| Freq. sees wild boar | Farmer interview | -0.02 | 0.98 | 0 | 1 |
| Freq. sees roe deer | Farmer interview | -0.24 | 0.83 | -0.56 | 0.60 |
| Freq. sees badger | Farmer interview | -0.08 | 0.94 | 1.34 | 0.26 |

Significant results are shown in bold type. a.s.l = above sea level.

to rule out the presence of *Mycobacterium avium* subspecies [28]. The analytical specificity of the PCR was tested on 112 DNA samples from pure cultures from 28 *Mycobacterium* species (not belonging to MTC), 5 *Streptomyces sp.*, and one *Lactobacillus brevis* strain. Cross-reaction was detected in a few *M. avium* subsp. *hominissuis* isolates.

## Statistics

We decided to use only non-parametric statistics owing to the relatively low sample size (n = 24 cattle farms). The Chi square or Fisher's test were used (when appropriate) for categorical data, while Mann-Whitney's U test was employed for numerical data.

## Results

### The local MTC maintenance host community

The local host community is composed of 3,100 cattle, 309 goats and 4,786 sheep as regards livestock, and between 232 and 431 wild boar (mean density 2.3 to 4.3/km$^2$ based on photo-trapping and hunting data, respectively) and 91 badgers (photo-trapping; mean density 0.9/km$^2$) as regards suitable wildlife MTC host species.

Of the 24 cattle farms sampled, eight were negative to the skin test during 2012–2016. The remaining 16 farms had reactors to the skin test, with an average of 28.6 positive skin-test animals/year, and none of the reactor farms was consistently positive in all five years (the largest series of consecutive years that a farm was positive was three and four, and this occurred on only two of the 16 reactor farms). This produced a chessboard-like figure of TB incidence in

cattle in the study area (S1 Table). All the goats tested during the five previous years had obtained negative results to the intradermal skin test. However, TB compatible lesions were detected in four sheep as part of the slaughterhouse surveillance, revealing an unknown prevalence of infection in this host species.

With regard to wildlife, we sampled 57 hunter-harvested wild boar. The TB-compatible lesion prevalence was 54% (31/57), while the infection prevalence confirmed by culture was 8.8% (5/57). Only one road-killed badger was submitted for culture and tested positive. All of the cattle (N = 7), sheep (N = 4), wild boar (N = 5) and badger (N = 1) isolates obtained in the study were identified as *M. bovis* and were classified by means of SB0134 spoligotype (according to the Mbovis.org database).

## Farm TB status and environmental DNA

The results of the annual intradermal test were employed as a basis on which to classify the farms as those with a higher risk, those that were positive to TB for at least two years (consecutive or otherwise) (n = 10), and those with a lower risk, i.e., those with reactors in one year or without reactors (n = 14) in the period of study (2012–2016). Environmental MTC DNA was detected in cattle from 12 of the 24 farms sampled (50% positive farms; 15 positive samples out of 48, 31.3% positive samples). There was no significant association between the higher risk area and the detection of environmental DNA (Fisher's test, P = 0.982). *Mycobacterium avium* subspecies was detected in only one sample from a farm on which the additional sponge was positive only to MTC.

Regional pastures were used by 10 of the 12 MTC DNA positive farms but by only 3/12 of the MTC DNA negative ones (Fisher's test, P = 0.006. Fig 1). The use of municipal pastures was only marginally significant (Fisher's test, P = 0.088). With regard to the quantitative variables, altitude and distance to woodland were significantly related to MTC DNA detection (Mann-Whitney's U test, P = 0.014 and P = 0.006, respectively. Table 1). In other words, there is a greater risk of detecting MTC DNA on farms at higher altitudes and that are closer to wildlife habitats (Fig 2). In contrast, we found no significant difference between "higher risk" and "lower risk" cattle farms as regards using regional pastures or municipal pastures (Fisher's test, P = 0.9449 and P = 0.704, respectively), and no differences were found for the 20 quantitative variables. Farm altitude and distance to woodland proved significant (both factors are correlated with each other; Pearson's *r* = 0.45; P = 0.024). The use of regional pastures is also correlated with the distance to woodland, but not with altitude; ANOVA P = 0.021, P = 0.448, respectively).

## Discussion

The ED carried out in this tuberculosis hotspot region evidenced that MTC is maintained in the study area owing to a multi-host community composed of domestic and wild hosts (with confirmed infection in cattle, sheep, wild boar and badger), despite all the efforts made by the TB eradication program. Environmental sampling was used as an additional test to detect mycobacteria on the farms, and the results revealed that shared regional pastures and proximity to woodlands were risk factors.

About 40 individual cattle in the study area become infected annually (based on 2016 official TB testing). Considering that the sensitivity of the single intradermal test is around 70% [32], it is possible that there was a higher number of infected cattle (around 52) in that year in the study area. There were, simultaneously, also about 20 to 38 MTC infected wild boar (at a minimum prevalence of 8.8%), plus an unknown number of infected sheep and badgers, sharing the habitat with cattle. It is consequently necessary to assume that MTC, as a paradigmatic

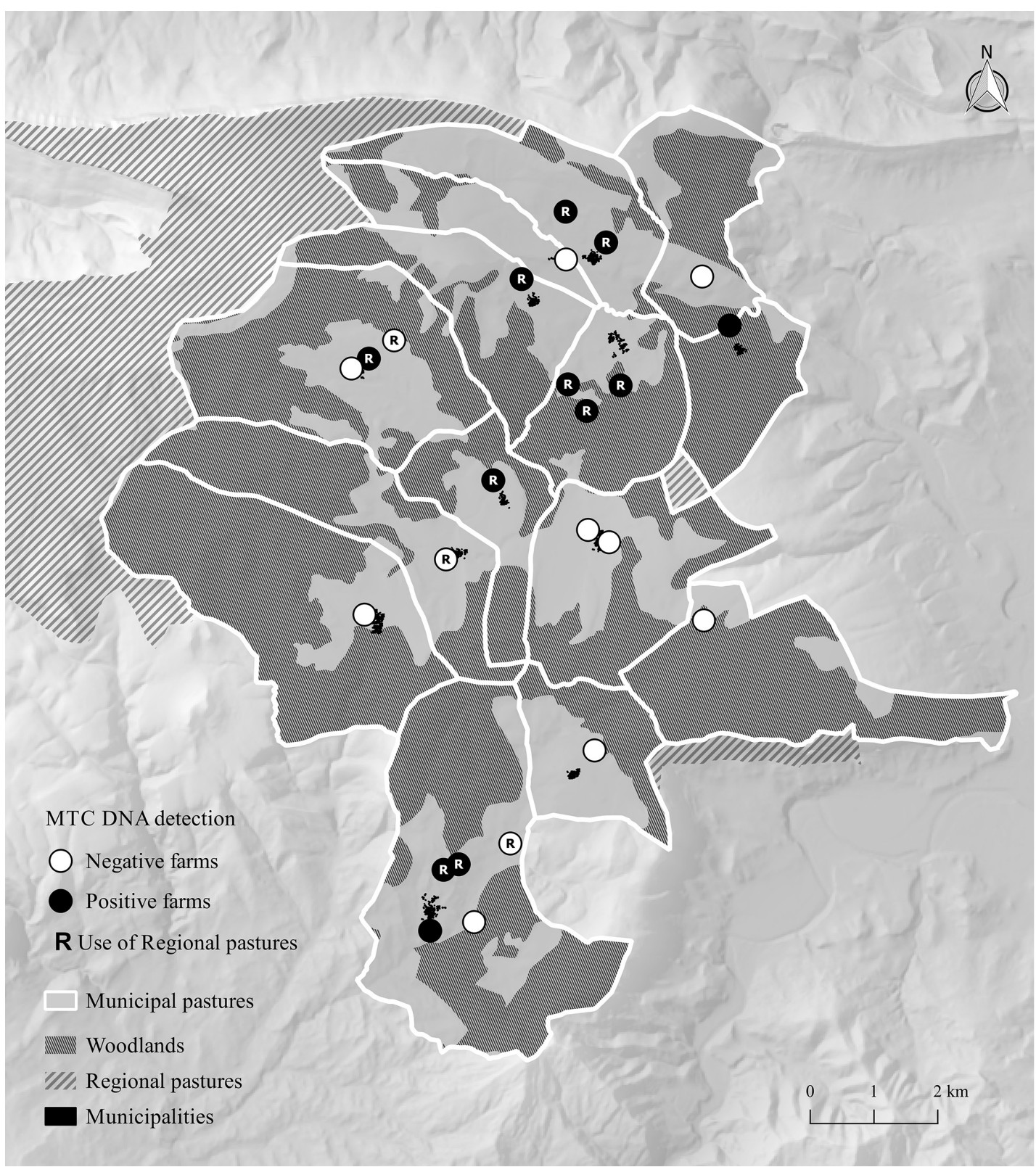

**Fig 1. Shared use of communal pastures as a risk factor depending on environmental MTC DNA detection.**

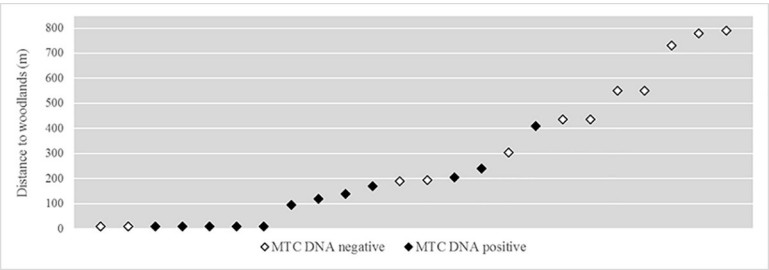

**Fig 2. Proximity of cattle shelters to woodlands as a risk factor, depending on environmental MTC DNA detection.** Points represent the different farms studied. Distance is expressed in meters.

multi-host infection, is maintained by a complex matrix of wild and domestic hosts. Any intervention should, therefore, target all possible hosts and not just cattle. Moreover, host populations are not static. The number of wild boar in the study region has, for example, tripled since 1990, and continues to increase [33,34]. Badger abundance is also known to have increased in northern Spain [35]. This and other changes in the composition of the host community, along with the changes in wildlife and livestock management and in livestock numbers, will likely modify the relative importance of the risk factors for shared infections over time.

This study suggests that on-animal environmental DNA sampling can help when assessing contact risk as regards MTC in livestock at the herd level. Regional pastures can be freely used by all herds of cattle in this region, while municipal pastures are shared only with farms located in the same municipality. One risk linked to shared pastures is obviously contact with other herds of cattle. However, additional risks as regards regional (or municipal) pastures are the contact with other livestock (in this case, flocks of sheep) and the contact with wildlife. Regional pastures tend to be at higher altitudes and are more suitable for wildlife such as wild boar. In our specific study area, the regional pasture area is, moreover, a protected natural park, where wild boar hunting is restricted. The identification of communal pastures as a risk factor for cattle TB coincides with recent findings obtained from a large-scale survey carried out in Castilla y León, Spain, in which communal pastures were also found to be a risk factor for wild boar TB [7]. We suggest that these factors are a proxy for the risk of contact with wildlife MTC hosts, such as badgers and wild boar. Epidemiological links between badgers, wild boar and cattle have already been evidenced in northern Spain [9, 36] and elsewhere [37].

Our results do not allow us to reject the initial hypothesis: the detection of environmental DNA in cattle could contribute to the identification of risk factors at the cattle herd level and, always in combination with the official TB tests of individual animals, could represent a useful epidemiological tool with which to investigate MTC maintenance in TB hotspot areas. This is of particular relevance given the difficulties involved in interpreting TB testing outcomes in low-prevalence settings [38]. While this finding might, at first glance, appear to be counter-intuitive, it can easily be explained: a positive official intradermal test reflects that cattle have previously been exposed to MTC, and the reactors are almost immediately slaughtered and hence removed from the study herds. In contrast, MTC DNA detection in a herd of cattle, while not necessarily reflecting infection, suggests current contact with an MTC-contaminated source, e.g. wildlife or the environment. The low prevalence and incidence in the area and the apparently random detection of positive tests could suggest an environmental origin of the new cases. Although this study was not able to elucidate the true origin of MTC infection in the herds of cattle studied, MTC DNA detection has made it possible to reveal specific risk factors that could be dealt with by means of farm biosafety.

Additional research is required to verify the usefulness of this technique in different situations in which its implementation has the potential to support the economic and sanitary viability of livestock farms on, for example, farms with recurrent cases of TB, or after it has been stamped out and the farm disinfected in order to avoid reinfection from environmental sources. We are also of the opinion that soil sampling could be a very valuable complementary tool that would enable us to obtain information that could be contrasted with our data, thus allowing a more complete view of the epidemiological scenario of the region studied and permitting us to obtain better conclusions. It is, therefore, necessary to carry out additional research into which tools other than those used in this study could be integrated, thus complementing the information required to carry out comprehensive and useful epidemiological diagnoses.

Additional care is advised for two reasons. First, the real time PCR used to detect MTC DNA is based on the IS*6110* insertion sequence. While all efforts have been made to confirm specificity, such as ruling out the presence of *Mycovacterium avium* subspecies, a small risk of cross reaction with some environmental mycobacteria or with other agents, such as certain *Streptomyces* strains, could exist. We assume this limited specificity in some scenarios, although this should not invalidate our results. Moreover, *Mycobacterium microti*, a pathogen belonging to the MTC but not included among the main causal agents of bovine TB, will cause positive PCR results [39]. Fortunately, *M. microti* is very rarely reported in Spain (only five cases are recorded in mycodb.es) [40].

The tool described herein, which detects MTC DNA at the herd level, opens up new avenues of epidemiological research and facilitates ED. This tool will hopefully provide additional insights into the herd-level risk factors as regards contact with MTC and other pathogens in a broad range of settings.

## Supporting information

**S1 Fig. Location of the photo-trapping grids in the study area.**
(TIF)

**S1 Table. Official annual results as regards TB testing on the farms sampled.** Each column shows the positive animals (cattle) in the respective years, categorized by diagnostic technique according to the following color code: Positive Tuberculin Intradermal Reaction (red); Positive Interferon-γ Gamma Release Assay (blue); Positive Slaughter (Lesions) (yellow); Positive Culture (green). If the same animal tested positive for more than one diagnostic test, only the color of the most restrictive technique is shown. The last column shows the environmental MTC DNA sampling results, categorized as MTC (*Mycobacterium tuberculosis* Complex), MAC (*Mycobacterium avium* Complex), *Mycobacterium* sp. (genus *Mycobacterium* not belonging to MTC or MAC) and 0 (negative or not conclusive).
(PDF)

## Acknowledgments

We are indebted to all the farmers who have participated in this study. We are also grateful to all the hunters and field veterinarians who diligently collaborated in sample collection. Last, but by no means least, we would also like to thank the Governments of Navarra's Environmental Service and Forest Guard Section their full collaboration in this project.

## Author Contributions

**Conceptualization:** Cesar Fernández, Lucas Domínguez, Christian Gortázar.

**Data curation:** Jordi Martínez-Guijosa, Miguel Veiga, Javier Ferreres.

**Formal analysis:** Jordi Martínez-Guijosa, Miguel Veiga, Javier Ferreres.

**Investigation:** Jordi Martínez-Guijosa, Beatriz Romero, José Antonio Infantes-Lorenzo, Elena Díez, Mariana Boadella, Ana Balseiro, David Navarro, Inmaculada Moreno, Mercedes Domínguez, Cesar Fernández.

**Methodology:** Beatriz Romero, Lucas Domínguez.

**Project administration:** Lucas Domínguez, Christian Gortázar.

**Resources:** Beatriz Romero, Elena Díez, David Navarro, Cesar Fernández.

**Supervision:** Mariana Boadella, Cesar Fernández.

**Validation:** Beatriz Romero.

**Visualization:** Jordi Martínez-Guijosa, Miguel Veiga, Christian Gortázar.

**Writing – original draft:** Jordi Martínez-Guijosa.

**Writing – review & editing:** Jordi Martínez-Guijosa, Beatriz Romero, Christian Gortázar.

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
