## [Decision Letter · Decision Letter 0]

10 Oct 2019

PONE-D-19-20068

Environmental DNA points at shared pastures and proximity to woodlands as tuberculosis risk factors for cattle in multi-host settings

PLOS ONE

Dear Mr. Martinez-Guijosa,

Thank you for submitting your manuscript to PLOS ONE. After careful consideration, we feel that it has merit but does not fully meet PLOS ONE’s publication criteria as it currently stands. Therefore, we invite you to submit a revised version of the manuscript that addresses the points raised during the review process.

As you will see the Referee 1# was very positive (minor revisions) about the quality of you work whereas the Referee 2# supported the rejection of your manuscript. In these situations, I prefer to recommend to the authors to pay attention to all points raised by the most sceptic reviewer (he mainly argued against the interpretation of your results) and even to perform extra laboratory work and the re-analysis of the data to prevent rejection in a second round. Please, take this negative as an opportunity to improve your research.

We would appreciate receiving your revised manuscript by Nov 24 2019 11:59PM. To enhance the reproducibility of your results, we recommend that if applicable you deposit your laboratory protocols in protocols.io, where a protocol can be assigned its own identifier (DOI) such that it can be cited independently in the future. For instructions see: http://journals.plos.org/plosone/s/submission-guidelines#loc-laboratory-protocols

We look forward to receiving your revised manuscript.

Kind regards,

Emmanuel Serrano, PhD

Academic Editor

PLOS ONE

**Journal Requirements:**

"I have read the journal's policy and the authors of this manuscript have the following competing interests: MB and SABIOTEC are consultants and offer risk assessment at the wildlife-livestock interface."

3. We note that  Figure(s) 1 in your submission contain [map/satellite] images which may be copyrighted. All PLOS content is published under the Creative Commons Attribution License (CC BY 4.0), which means that the manuscript, images, and Supporting Information files will be freely available online, and any third party is permitted to access, download, copy, distribute, and use these materials in any way, even commercially, with proper attribution. For these reasons, we cannot publish previously copyrighted maps or satellite images created using proprietary data, such as Google software (Google Maps, Street View, and Earth). For more information, see our copyright guidelines: http://journals.plos.org/plosone/s/licenses-and-copyright.

a) You may seek permission from the original copyright holder of Figure(s) [#] to publish the content specifically under the CC BY 4.0 license.  

**Comments to the Author**

1. Is the manuscript technically sound, and do the data support the conclusions?

Reviewer #1: Yes

Reviewer #2: Partly

2. Has the statistical analysis been performed appropriately and rigorously? 

Reviewer #1: I Don't Know

Reviewer #2: No

3. Have the authors made all data underlying the findings in their manuscript fully available?

Reviewer #1: Yes

Reviewer #2: Yes

4. Is the manuscript presented in an intelligible fashion and written in standard English?

Reviewer #1: Yes

Reviewer #2: Yes

5. Review Comments to the Author

Reviewer #1: Here are some comments about your work that I would like you to consider:

Line number (LN) 3: The authors must write accent mark in: Martínez-Guijosa

In the abstract: The authors must remove the abbreviations: TB (tuberculosis) and ED (epidemiological diagnosis)

LN42: The authors must write: Mycobacterium instead of M.

LN80: The authors must write: Materials (in plural)

LN103: The authors must change: …described by Rowcliffe et al. 2008 [21], for: …described previously [21]

LN100-110: The authors must indicate the study period (2012-2016 also or only 2015/2016? Not specified)

LN110: The authors must write: and submitted for bacteriological culture and molecular identification in a...

LN115: The authors must clarify how many cattle (since if there were a total of 3,100 cattle in 36 herds, how many cattle were in the 24 herds were studied?) and goats (a total of 309 goats?) are tested for TB.

LN116 and LN117: The authors must write: Spanish Eradication Program of Bovine Tuberculosis just like they have written in LN127.

LN117: The authors must clarify if in Spanish Eradication Program of Bovine Tuberculosis (year 2018) were available the data for cattle and goats for the period 2012-2016 or just the cattle’s data

In the Table 1: The authors should analyze if the expression “Freq. sees roe deer” refers to seeing roe deer (Capreolus capreolus) or actually they mean red deer (Cervus elaphus). If it is not a spelling mistake, the authors must make reference in the introduction (LN49-LN54) to the relationship between roe deer and tuberculosis, e. g. “the roe deer (Capreolus capreolus) are not a significant TB reservoir” (Balseiro A, Oleaga Á, Orusa R, Robetto S, Domenis L, Zoppi S, et al. Tuberculosis in roe deer from Spain and Italy. Vet Rec. 2009;164: 468-470). “However, once infected, it develops lesions reflecting a clear ability for bacterial excretion and therefore transmission to other species, most likely by indirect contact. It could thus be a spillover host included in a multi-host component reservoir in endemic areas. Consequently, passive surveillance is essential to detect infection and to implement specific management to limit interactions with cattle, where infected roe deer are found” (Lambert S, Hars J, Réveillaud E, Moyen JL, Gares H, Rambaud T, et al. Host status of wild roe deer in bovine tuberculosis endemic areas. Eur J Wildl Res. 2017;63: 15).

LN142 and LN143: The authors must clarify if on each farm, each of the two sponges were used on the back of a single animal (1 sponge / 1 animal) or the same sponge was scrubbed ten times on the back of the two randomly chosen cattle (1 sponge / 2 animals). In addition, the authors must explain why they chose the back and not another anatomical region of the animal such as the head or legs to take the samples of environmental DNA.

LN149: The authors must write the primers and probe used to detect IS6110 or add another reference along with the [25].

LN166: The authors must write: The local host community is composed of XXXX cattle, just like they have written in LN115.

LN180-LN182: The authors must indicate how many MTC isolates were obtained in the study in each animal species, e. g. “All cattle (N= XXX), sheep (N= 4), wild boar (N= 5) and badger (N= 1) isolates obtained in the study were identified as M. bovis and were classified with SB0134 spoligotype (according to the Mbovis.org database)”.

LN187-190: Authors should refer to the results observed in: “A conventional PCR was also used to discard the presence of Mycobacterium avium subspecies [26]”.

LN216 and LN217. The authors must explain where they obtained these data (“also about 23 to 43 MTC infected wild boar -at a minimum prevalence of 10%-”) because previously they indicated: “With regard to wildlife, we sampled 57 hunter-harvested wild boar. The TB-compatible lesion prevalence was 54% (31/57) and the infection prevalence confirmed by culture was 8.8% (5/57)”.

LN251-LN257: The authors must explain why the environmental DNA samples that proved to be positive for the presence of MTC were not analyzed by spoligotyping as did Barbier el at. 2016 [26], especially considering that “all cattle, sheep, wild boar and badger isolates obtained in the study were identified as M. bovis and were classified with SB0134 spoligotype”.

LN272-LN274: The authors must write: Gortázar C, Che Amat A, O’Brien DJ. Open questions and recent advances in the control of a multi-host infectious disease: Animal tuberculosis. Mamm Rev. 2015;45: 160–175. doi:10.1111/mam.12042

LN277: Authors should change the URL and put: https://www.mapa.gob.es/es/ganaderia/temas/sanidad-animal-higiene-ganadera/pnetb_2018_tcm30-436761.pdf

LN278: The authors must write accent mark in: Liébana, Gómez-Mampaso, Galán

LN 283: The authors must write: infections (no italic letter)

LN 296: The authors must delete: Khudyakov YE, editor.

LN297: The authors must delete: –e71074

LN302 and LN303: The authors must delete: Fooks A, editor. – e18058

LN318 and LN319: Authors should change the URL and put:

https://www.mapa.gob.es/es/ganaderia/temas/sanidad-animal-higiene-ganadera/patubes2017_3_tcm30-378321.pdf

LN325 and LN326: The authors must write: Wildlife/Livestock Interface. Transbound Emerg Dis. 2017;64: 1148–1158. doi:10.1111/tbed.12480

LN331: The authors must write: Mycobacterium bovis (with italic letter)

LN347: The authors must delete: Inacio J, editor. –e88824

LN373: The authors must write accent mark in: Pérez A,

Reviewer #2: This is an interesting paper describing the relationship between on-animal environmental MTC DNA detection and bovine TB risk in a multi-host scenario. Detection of environmental MTC DNA in cattle herds is an interesting tool, however I think results obtained in this paper does not completely support conclusions and some important issues need to be considered for publication

The main concerns about this paper are:

- Interpretation of results: Authors conclude that on-animal MTC DNA detection could be a good proxy for TB risk in cattle herds and that it could be used as an epidemiological tool to understand unexpected results.

To support this conclusion it could be expected an association between the detection of MTC DNA and the presence of new TB-positive animal at heard level. However, this hypothesis has not been assessed in this work. Furthermore, there is not association between herds where MTC DNA was detected and herd with higher risk of TB.

MTC-DNA detection in cattle herds was related to the use of communal pasture, altitude and distance to woodland, but it does not mean that this technique was related with the risk of TB infection. Other risk factors also related with TB risk like the abundance of wild boar or the presence of risk points, seem not to be associated with MTC-DNA detection, and it should be discussed.

- Specificity of the technique: In discussion section the authors assume that the PCR technique used in this work could have small risk of cross reaction with some environmental mycobacteria or with other agents. But, how small is this risk in an scenario where environmental mycobacteria have been isolated repeatedly isolated from cattle and wildlife? Could the specificity be improved using confirmative PCR assays based on exclusive M. bovis gene?

Furthermore, about the sensitivity, could the absence of MTC DNA be ensured testing only two animals per herd?

- Cost-benefits of on-animal MTC detection: Authors should discuss whether this technique is appropriate regarding to cost-benefits criteria.

Other issues about the paper are:

- Lines 178-179: Although the prevalence of tb-like lesions in wild boar was a 54%, the infection only was confirmed in a 8.8%. I think this issue should be discussed as an unexpected result. Could infections with other agents related with MTC produce this kind of lesion?

- Lines 184-486: Which scientific criteria have been used to stablish hard or low TB risk herds? Perhaps, low TB risk herds could be those without positive animals. Furthermore, in order to avoid theoretical classification, quantitative variable would be used to measure TB risk (e.g. number of positive animals or mean prevalence in the last 10 years, etc.)

- Lines 188-190: When were samplings to detect MTC environmental DNA carried out (simultaneously, in different years…). Results of MTC samplings should be added to s1 table.

- Lines 193-194: Do cattle herds sharing the same municipal pasture really coincide as regards their MTC DNA detection status? I think it should be clarify. In Figure 1 I count five municipalities with more than one cattle herds and only in three of them I can see a complete coincidence in MTC status.

- Lines 195-196. Are there any correlation between altitude and woodland distance? I could be an interesting point to assess.

- Line 233. Was the use of communal pastures really identified as a risk factor for TB in cattle in this work?

6. PLOS authors have the option to publish the peer review history of their article (what does this mean?). If published, this will include your full peer review and any attached files.

Reviewer #1: Yes: José Manuel Benítez-Medina

Reviewer #2: No

---

## [Author Response · Author response to Decision Letter 0]

23 Nov 2019

PONE-D-19-20068

Environmental DNA points at shared pastures and proximity to woodlands as tuberculosis risk factors for cattle in multi-host settings

Comments to the Author

1. Is the manuscript technically sound, and do the data support the conclusions?

Reviewer #1: Yes

Reviewer #2: Partly

2. Has the statistical analysis been performed appropriately and rigorously? 

Reviewer #1: I Don't Know

Reviewer #2: No

3. Have the authors made all data underlying the findings in their manuscript fully available?

Reviewer #1: Yes

Reviewer #2: Yes

4. Is the manuscript presented in an intelligible fashion and written in standard English?

Reviewer #1: Yes

Reviewer #2: Yes

5. Review Comments to the Author

Reviewer #1: Here are some comments about your work that I would like you to consider:

Line number (LN) 3: The authors must write accent mark in: Martínez-Guijosa

->OK, done.

In the abstract: The authors must remove the abbreviations: TB (tuberculosis) and ED (epidemiological diagnosis)

->OK, done.

LN42: The authors must write: Mycobacterium instead of M.

->OK, done.

LN80: The authors must write: Materials (in plural)

->OK, done.

LN103: The authors must change: …described by Rowcliffe et al. 2008 [21], for: …described previously [21]

->OK, done.

LN100-110: The authors must indicate the study period (2012-2016 also or only 2015/2016? Not specified)

->OK, done. In lines 102-103 is stated “installed from September 5 to October 12 (2016)” to specify the camera trapping study period. In line 131 is stated “All the work performed on the farms was carried out between September 2016 and January 2017” to specify on-farm study period.

LN110: The authors must write: and submitted for bacteriological culture and molecular identification in a...

->OK, done.

LN115: The authors must clarify how many cattle (since if there were a total of 3,100 cattle in 36 herds, how many cattle were in the 24 herds were studied?) and goats (a total of 309 goats?) are tested for TB.

->Thank you for the comment. In lines 118-119 is stated: “The 24 herds selected for the study added 2,786 cattle (89.87% of the total cattle in the study area)”, to specify how many cattle were in the 24 selected herds. 

Yes, a total of 309 goats were tested for TB in the year 2016, since “All cattle and goats are tested for TB on an annual basis by means of the official single intradermal tuberculin test according to the Spanish bovine tuberculosis eradication program…”.

LN116 and LN117: The authors must write: Spanish Eradication Program of Bovine Tuberculosis just like they have written in LN127.

->OK, done.

LN117: The authors must clarify if in Spanish Eradication Program of Bovine Tuberculosis (year 2018) were available the data for cattle and goats for the period 2012-2016 or just the cattle’s data.

->OK, done. Thank you for the comment. The Spanish Eradication Program of Bovine Tuberculosis explains the protocols and procedures followed in Spain to achieve the eradication of Bovine Tuberculosis in cattle and goats with contact with cattle (directly or through common pastures) or those flocks considered as a possible source of infection to cattle. Information about the prevalence of the disease in cattle herds at national, provincial and regional levels are given in this annual report, including the period of this study (2012-2016). Since prevalence and incidence of TB in goats is not included in this Spanish program, the Government of Navarra, which actively participated in the conduct of this study, provided them.

In line 121 is stated: “…and data for the period 2012-2016 was, therefore, kindly provided by the Government of Navarra” to clarify this fact.

In the Table 1: The authors should analyse if the expression “Freq. sees roe deer” refers to seeing roe deer (Capreolus capreolus) or actually they mean red deer (Cervus elaphus). If it is not a spelling mistake, the authors must make reference in the introduction (LN49-LN54) to the relationship between roe deer and tuberculosis, e. g. “the roe deer (Capreolus capreolus) are not a significant TB reservoir” (Balseiro A, Oleaga Á, Orusa R, Robetto S, Domenis L, Zoppi S, et al. Tuberculosis in roe deer from Spain and Italy. Vet Rec. 2009;164: 468-470). “However, once infected, it develops lesions reflecting a clear ability for bacterial excretion and therefore transmission to other species, most likely by indirect contact. It could thus be a spillover host included in a multi-host component reservoir in endemic areas. Consequently, passive surveillance is essential to detect infection and to implement specific management to limit interactions with cattle, where infected roe deer are found” (Lambert S, Hars J, Réveillaud E, Moyen JL, Gares H, Rambaud T, et al. Host status of wild roe deer in bovine tuberculosis endemic areas. Eur J Wildl Res. 2017;63: 15).

->OK, done. Thank you for the comment and the useful recommendations. It is not a spelling mistake, it refers to Capreolus capreolus sights, since Cervus elaphus in completely absent in the study area.

In lines 49-54, the reference to wild TB reservoirs in Spain is only briefly explained at geographical level. However reference [7] (Gortázar C, Fernández-Calle LM, Collazos-Martínez JA, Mínguez-González O, Acevedo P. Animal tuberculosis maintenance at low abundance of suitable wildlife reservoir hosts: A case study in northern Spain. Prev Vet Med. 2017;146: 150–157) include roe deer in its sampling efforts.

In lines 55-66 the role of wild reservoirs in the Atlantic Spain was explained at species level, so the following sentence is now added in lines 63-65 to clarify the potential role of roe deer in TB transmission: “Roe deer (Capreolus capreolus), is an abundant cervid in northern Spain [14], and despite cases of infected roe deer have been evidenced, it does not appear to be a significant reservoir of TB [15,16]”. Suggested references included. 

LN142 and LN143: The authors must clarify if on each farm, each of the two sponges were used on the back of a single animal (1 sponge / 1 animal) or the same sponge was scrubbed ten times on the back of the two randomly chosen cattle (1 sponge / 2 animals). In addition, the authors must explain why they chose the back and not another anatomical region of the animal such as the head or legs to take the samples of environmental DNA.

->Thank you for the comment. Two sponges were used per farm and in two different animals. Therefore, two animals provide information about the environmental DNA sampling. It has been included in line146-147 “On each farm, two sponges were scrubbed ten times on the scapula regions of two randomly chosen cattle (one sponge by animal). The scapula region, and no other anatomical region of the animal, was chosen because it is the most accessible area during cattle handling which is relatively close to the brisket, one of the preferred areas for sampling pathogens on cattle hides [28]. In addition, it is a large and quasi-horizontal area, where dust particles can easily deposit.”

LN149: The authors must write the primers and probe used to detect IS6110 or add another reference along with the [25].

->Ok, done. Information about the sequences of primers and probe has been included in the manuscript in lines 155-157 as follows: “A real time PCR with which to detect IS6110 was used to detect MTC in the samples, using the primers (forward 5’- GGTAGCAGACCTCACCTATGTGT-3’; reverse 5’-AGGCGTCGGTGACAAAGG-3’) and probe (5’ FAM-CACGTAGGCGAACCC-MGB-NFQ 3’) described previously [30].

LN166: The authors must write: The local host community is composed of XXXX cattle, just like they have written in LN115.

->OK, done.

LN180-LN182: The authors must indicate how many MTC isolates were obtained in the study in each animal species, e. g. “All cattle (N= XXX), sheep (N= 4), wild boar (N= 5) and badger (N= 1) isolates obtained in the study were identified as M. bovis and were classified with SB0134 spoligotype (according to the Mbovis.org database)”.

->OK, done.

LN187-190: Authors should refer to the results observed in: “A conventional PCR was also used to discard the presence of Mycobacterium avium subspecies [26]”.

->Thank you for this comment. Only one positive results to Mycobacterium avium complex was detected. It was included in the manuscript in lines 198-199: “Mycobacterium avium subspecies was only detected in one sample from a farm where the additional sponge was only positive to MTC”.

LN216 and LN217. The authors must explain where they obtained these data (“also about 23 to 43 MTC infected wild boar -at a minimum prevalence of 10%-”) because previously they indicated: “With regard to wildlife, we sampled 57 hunter-harvested wild boar. The TB-compatible lesion prevalence was 54% (31/57) and the infection prevalence confirmed by culture was 8.8% (5/57)”.

->Thank you for the comment. 10% was selected as an approximate minimum TB prevalence in the study area to exemplify the potential numbers of infected wild boars in that area, because of simplicity. Nevertheless, is true that, to be more accurate, we could use the exact percentage presented in our results. The numbers have been recalculated with this 8.8% to solve this problem (lines 224-225: “…20 to 38 MTC infected wild boar …”).

LN251-LN257: The authors must explain why the environmental DNA samples that proved to be positive for the presence of MTC were not analyzed by spoligotyping as did Barbier el at. 2016 [26], especially considering that “all cattle, sheep, wild boar and badger isolates obtained in the study were identified as M. bovis and were classified with SB0134 spoligotype”.

->Thank you for your comments. Barbier et al. used the 6 positive samples of their study. They could only provide a complete profile in one of the samples, 3 profiles were not completed and the 2 samples left were negative. They did not provide information about the Ct values of the samples at the real time PCR used. In our hands, the Ct values obtained for most of the samples were > 35 and it cannot allow the minimum quantity and quality of DNA for spoligotyping purposes. 

LN272-LN274: The authors must write: Gortázar C, Che Amat A, O’Brien DJ. Open questions and recent advances in the control of a multi-host infectious disease: Animal tuberculosis. Mamm Rev. 2015;45: 160–175. doi:10.1111/mam.12042

->OK, done.

LN277: Authors should change the URL and put: https://www.mapa.gob.es/es/ganaderia/temas/sanidad-animal-higiene-ganadera/pnetb_2018_tcm30-436761.pdf

->OK, done.

LN278: The authors must write accent mark in: Liébana, Gómez-Mampaso, Galán

->OK, done.

LN 283: The authors must write: infections (no italic letter)

->OK, done.

LN 296: The authors must delete: Khudyakov YE, editor.

->OK, done.

LN297: The authors must delete: –e71074

->OK, done.

LN302 and LN303: The authors must delete: Fooks A, editor. – e18058

->OK, done.

LN318 and LN319: Authors should change the URL and put:

https://www.mapa.gob.es/es/ganaderia/temas/sanidad-animal-higiene-ganadera/patubes2017_3_tcm30-378321.pdf

->OK, done.

LN325 and LN326: The authors must write: Wildlife/Livestock Interface. Transbound Emerg Dis. 2017;64: 1148–1158. doi:10.1111/tbed.12480

->OK, done.

LN331: The authors must write: Mycobacterium bovis (with italic letter)

->OK, done.

LN347: The authors must delete: Inacio J, editor. –e88824

->OK, done.

LN373: The authors must write accent mark in: Pérez A,

->OK, done.

Reviewer #2: This is an interesting paper describing the relationship between on-animal environmental MTC DNA detection and bovine TB risk in a multi-host scenario. Detection of environmental MTC DNA in cattle herds is an interesting tool, however I think results obtained in this paper does not completely support conclusions and some important issues need to be considered for publication

The main concerns about this paper are:

- Interpretation of results: Authors conclude that on-animal MTC DNA detection could be a good proxy for TB risk in cattle herds and that it could be used as an epidemiological tool to understand unexpected results.

To support this conclusion it could be expected an association between the detection of MTC DNA and the presence of new TB-positive animal at heard level. However, this hypothesis has not been assessed in this work. Furthermore, there is not association between herds where MTC DNA was detected and herd with higher risk of TB.

MTC-DNA detection in cattle herds was related to the use of communal pasture, altitude and distance to woodland, but it does not mean that this technique was related with the risk of TB infection. Other risk factors also related with TB risk like the abundance of wild boar or the presence of risk points, seem not to be associated with MTC-DNA detection, and it should be discussed.

->Thank you for the comment, we share your concern. In this study, what we propose is that this tool can be used as an indicator of the risk of environmental MTC DNA exposure, and by extension, it has the potential to indicate the risk of exposure to TB-causative agents, since these belong to the MTC. However, the possible interference with non-TB-causing MTC members is possible, and therefore we have maintained prudence in our discussion, referring at all times to the risk of exposure to MTC DNA, as well as the circumstantial utility of this tool, which must still be tested in different scenarios to determine its true potential to help in epidemiological diagnoses and to solve sanitary problems at the landscape or herd level. Even so, we firmly rely on the potential of this tool, since, in this specific case, it has helped to identify certain risk factors in which authorities and farmers can work and stay alert. Well informed about the scope of our results and their limitations, the stakeholders involved in the problem have information with a scientific basis to try to solve their problems, which, with the information they had so far, could not solve despite their professionalism and their strictness.

On the other hand, when we state that this tool can help to understand unexpected results, we do not mean that it can help the individual diagnosis of animals, but, as is the case, can help to interpret results that, only with Official data are hardly interpretable, and therefore, they give no clues to develop specific actions to solve the problem.

In this sense, the lack of association between the detection of MTC DNA and the presence of new TB-positive animal at herd level can be due to various factors. First, it may be conditioned by the sensitivity problems of the diagnostic tests used (de la Rua-Domenech et al., 2006), especially taking into account the low incidence of TB in the area, and, second, it may be due to limitations (previously discussed) of the environmental DNA detection tool used in this study. However, given the impossibility of unravelling the reason for this issue, we believe that this lack of association shows how the use of new epidemiological tools that provide complementary data to existing epidemiological information can help solve specific problems in complex epidemiological scenarios.

Regarding the “abundance” of wild boar, in the first place we do not know exactly the role that this species plays in this environment, so it is possible that its potential as a reservoir and transmitter of the disease is associated with specific areas (such as the regional pastures of the Sierra de Urbasa-Andía, where hunting is not allowed), or the access of these animals to specific areas of the facilities during the winter, thus confounding their effect with the significant variables related with the use of regional pastures or the distance of the facilities to the forest, respectively. 

Secondly, it is important to remark that it is not “abundance” of wild boar, but the perception of each farmer of the presence of the species. This measure is individual and subjective, and, despite useful in some specific cases, these particular discrepancies can invalidate the potential effect of this variable in this particular case.

As for the risk points, the number of potential interaction risk points does not have to imply a greater risk of exposure to livestock, since, depending on the nature of the point and the use made by the animals of the different species potentially involved in the epidemiological system, accessibility to a single risky point may pose a greater risk than access to several of them.

- Specificity of the technique: In discussion section the authors assume that the PCR technique used in this work could have small risk of cross reaction with some environmental mycobacteria or with other agents. But, how small is this risk in an scenario where environmental mycobacteria have been isolated repeatedly isolated from cattle and wildlife? Could the specificity be improved using confirmative PCR assays based on exclusive M. bovis gene?

Furthermore, about the sensitivity, could the absence of MTC DNA be ensured testing only two animals per herd?

->Thank you for the comment.

1. Up to know, we cannot quantify the risk of cross reaction with some environmental mycobacteria.

2. The idea to use an additional real time PCR to confirm the previous result was evaluated. However, in most cases, the Ct values of the IS6110 PCR was high (Ct>35) and, taking into account that some M. bovis has several copies of the IS element, the sensitivity of another real time PCR could be lower than the IS6110. However, we did not include in this study the confirmatory PCR so no data are available.

3. No. Since 24 farms were studied, due to logistic reasons, and with the scientific objective of testing this new methodology to detect mycobacteria in the field, we decided to include only two sponges per farm, and see the results. We cannot assure that the absence of detection of environmental DNA in two samples represents the situation in the farm, but based on the results, it is a proxy or estimation of the farm MTC exposure risk.

- Cost-benefits of on-animal MTC detection: Authors should discuss whether this technique is appropriate regarding to cost-benefits criteria.

->Thank you for the comment. This methodology could be used in different scenarios, but it could be useful, for example, when: 1) recurrent TB is present in a farm or 2) after stamping out and disinfection of the farm to avoid reinfection of animals from environmental sources. Cost-benefit balance when using this technique will depend on the particular situation of the farm, and the balance will be positive in cases where the persistence of mycobacteria represents a problem for the economic and/or sanitary viability of the farm. However, it is not possible to establish a confident calculation of the generalized or commercial implementation of this technique, since now, the actual potential of its use in various scenarios where it could also be useful is unknown.

To discuss this issue, we added the following paragraph in lines 259-263: “Additional research is needed to verify the usefulness of this novel technique in different situations where its implementation has the potential to support the economic and sanitary viability of livestock farms, such as, for example, in farms with recurrent TB cases, or after stamping out and disinfection of the farm in order avoid animal reinfection from environmental sources.”

Other issues about the paper are:

- Lines 178-179: Although the prevalence of tb-like lesions in wild boar was a 54%, the infection only was confirmed in a 8.8%. I think this issue should be discussed as an unexpected result. Could infections with other agents related with MTC produce this kind of lesion?

->Thank you for the comment. Yes, as explained in Bollo et al. 2000, this kind of lesions can be produced by other mycobacteria (i.e. M. avium complex), or by other pathogens, such as Rhodococcus equi. Additionally, it was previously reported that the use of a single diagnosis post-mortem test on hunter-harvested wild boar, can underestimate the true prevalence, and that despite bacteriological culture is the reference test for TB diagnosis, it can generate false-negative results, and this should be considered when interpreting data (Santos et al. 2010). Nevertheless, we were aware of the possible biases in the prevalence obtained by any method, and therefore we considered the most conservative value as the reference value for later discussion, that is, the one based on culture.

Regarding the discussion of the nature of these results, it is not the objective of this study, since the prevalence in wild boar is provided as descriptive data to provide the reader with general information on the epidemiological context of TB in this area.

- Lines 184-486: Which scientific criteria have been used to stablish hard or low TB risk herds? Perhaps, low TB risk herds could be those without positive animals. Furthermore, in order to avoid theoretical classification, quantitative variable would be used to measure TB risk (e.g. number of positive animals or mean prevalence in the last 10 years, etc.)

->Thank you for the comment. As stated in lines 193-194: “…the farms were classified as farms with a higher risk, those that were positive to TB for at least two years (consecutive or otherwise) (n=10), and farms with a lower risk, i.e., those with reactors in one year or without reactors (n = 14), in the period of study (2012-2016).”

Since incidence and prevalence was generally very low in all herds (actually Navarra is a low prevalence region compared to other Spanish regions), and the temporal pattern of herd prevalence was apparently random, we decided to establish a criterion based on the positivity at herd level within the last five years (based on the “Historical persistence” described by Martínez-López et al. 2014 as one of the main factors for TB occurrence and persistence), rather than focusing on the number of positive animals in the herd.

- Lines 188-190: When were samplings to detect MTC environmental DNA carried out (simultaneously, in different years…). Results of MTC samplings should be added to s1 table.

->Thank you for the comment. The following state has been added in line 144 in order to clarify this point: “During January 2017, samples of environmental DNA were taken from cattle from 24 farms.”

Results of MTC samplings now added to S1 table and S1 table caption has been modified.

- Lines 193-194: Do cattle herds sharing the same municipal pasture really coincide as regards their MTC DNA detection status? I think it should be clarify. In Figure 1 I count five municipalities with more than one cattle herds and only in three of them I can see a complete coincidence in MTC status.

->Thank you for the comment. No, Fisher’s test not reveal significant association between municipal pasture groups and MTC DNA detection status (P = 0.2346).

To avoid confusions, the sentence “However, Fig 1 suggests that herds of cattle sharing the same municipal pastures often coincide as regards their MTC DNA detection status” has been removed. 

- Lines 195-196. Are there any correlation between altitude and woodland distance? I could be an interesting point to assess.

->Thank you for the comment. Yes, there is correlation between altitude and woodland distance (Pearson's r = 0.45; P= 0.02402).

The following sentence has been modified in lines 243-244 to clarify this question: “In addition, farm altitude and distance to woodland proved significant (both factors are correlated with each other; Pearson's r = 0.45; P= 0.02402). We suggest that closeness to woodland (and in this case, altitude) is a proxy for the risk of contact with wildlife MTC hosts such as badgers and wild boar.”

- Line 233. Was the use of communal pastures really identified as a risk factor for TB in cattle in this work?

->Thank you for the comment. No, only one kind of communal pastures (regional pasture) was identified as a risk factor. In lines 234-240, the word “communal” was used erroneously in reference to the specific regional pasture in the study area. It has been addressed in lines 234, 237, 238 and 240. The use of “communal” in lines 241 and 242 is correct, since it refers to communal pastures in a general sense, as pastures shared by several farmers.

6. PLOS authors have the option to publish the peer review history of their article (what does this mean?). If published, this will include your full peer review and any attached files.

Do you want your identity to be public for this peer review? For information about this choice, including consent withdrawal, please see our Privacy Policy.

Reviewer #1: Yes: José Manuel Benítez-Medina

Reviewer #2: No

---

## [Decision Letter · Decision Letter 1]

13 Feb 2020

PONE-D-19-20068R1

Environmental DNA points at shared pastures and proximity to woodlands as tuberculosis risk factors for cattle in multi-host settings

PLOS ONE

Dear Mr. Martinez-Guijosa,

Thank you for submitting your manuscript to PLOS ONE. After careful consideration, we feel that it has merit but does not fully meet PLOS ONE’s publication criteria as it currently stands. Therefore, we invite you to submit a revised version of the manuscript that addresses the points raised during the review process. In the second  review, a referee (wildlife vet; #2 reviewer) directly advised for the rejection of your work whereas the other (environmental microbiologist; #1 reviewer) asked for major revisions. Both reviewers thought that your work is timely and interesting but needs additional methodological clarifications. A third reviewer, however, was very positive and accepted your work for publication. In this situation, I usually reject manuscripts but in your case, because of the interest our your research, I prefer to give you a new chance to deal with these methodological issues. I strongly recommend paying particular attention to the questions raised by the #2 and #1 referee that will be invited again to revise your work.

We would appreciate receiving your revised manuscript by Mar 29 2020 11:59PM. To enhance the reproducibility of your results, we recommend that if applicable you deposit your laboratory protocols in protocols.io, where a protocol can be assigned its own identifier (DOI) such that it can be cited independently in the future. For instructions see: http://journals.plos.org/plosone/s/submission-guidelines#loc-laboratory-protocols

We look forward to receiving your revised manuscript.

Kind regards,

Emmanuel Serrano, PhD

Academic Editor

PLOS ONE

Reviewers' comments:

Reviewer's Responses to Questions

**Comments to the Author**

1. If the authors have adequately addressed your comments raised in a previous round of review and you feel that this manuscript is now acceptable for publication, you may indicate that here to bypass the “Comments to the Author” section, enter your conflict of interest statement in the “Confidential to Editor” section, and submit your "Accept" recommendation.

Reviewer #1: All comments have been addressed

Reviewer #2: (No Response)

Reviewer #3: (No Response)

2. Is the manuscript technically sound, and do the data support the conclusions?

Reviewer #1: Yes

Reviewer #2: Partly

Reviewer #3: Partly

3. Has the statistical analysis been performed appropriately and rigorously? 

Reviewer #1: I Don't Know

Reviewer #2: No

Reviewer #3: I Don't Know

4. Have the authors made all data underlying the findings in their manuscript fully available?

Reviewer #1: Yes

Reviewer #2: Yes

Reviewer #3: Yes

5. Is the manuscript presented in an intelligible fashion and written in standard English?

Reviewer #1: Yes

Reviewer #2: Yes

Reviewer #3: Yes

6. Review Comments to the Author

Reviewer #1: Thanks for including the suggestions. I believe that paper is now better understood. In any case, as you have clarified, this research is especially a methodological work that must be continued to clarify some issues and prove their true usefulness. Please review references 1 (2015; 45: 160–175.) and 3 (name bacteria in italics).

Reviewer #2: Dear authors,

Many thanks for responding all my comments included in the first round. I maintain that environmental MTC DNA detection is a promising and an interesting tool to be used in epidemiological studies. However, despite of modification included in this new version, data showed in the manuscript do not support suggestions and conclusions yet, and hence, I cannot recommend publication.

In this work, two criteria have been used to analyse TB status of the farms: TB testing results (based on official data between 2001 and 2016), and on-animal detection of environmental MTC DNA at herd level. Detection of MTC DNA was related with three epidemiological factors (probably correlated among them): altitude of the farm, distance to woodland and the use of regional pasture. However, TB risk level does not correlated with any of the risk factors proposed. In addition, no association was detected between TB risk level (based on data from 2001-2016) and environmental MTC DNA detection at herd level.

These results are prudently discussed in few sections of the paper. However, authors maintain in the manuscript some statements like “This study suggests that on-animal environmental DNA sampling can be a proxy for contact (or contact risk) with MTC in livestock, at the herd level”, “Our results allow us to accept the initial hypothesis: the detection of environmental DNA in cattle could contribute to risk factor identification at the cattle herd level and, always in combination with the official TB tests of individual animals, could represent a useful epidemiology tool to investigate MTC maintenance in TB hotspot areas” or “We consequently suggest that MTC DNA could be a good proxy for current infection risk. The low prevalence and incidence in the area and the apparently random detection of positive tests could indicate an environmental origin of the new cases”; which, from my point of view, are not supported by data showed in the manuscript.

To support these suggestions, it would be necessary an association between detection of MTC DNA and TB positivity at herd levels, but it has not been proven. I know that this fact can be due to the sensitivity problems of the diagnostic test used in low prevalence areas. But this fact could be also due to specificity problems of the MTC DNA detection, mainly in areas closed to woodlands where wild boar with a high percentage (26/31 (84%)) of microbiologically unconfirmed TB-like lesions (probably caused by closely related bacteria) are living.

Once validated, on-animal MTC DNA detection would be an useful tool to be implemented in low-prevalence areas like the studied one. However, in order to assess the association between environmental MTC DNA detection and the risk of TB appearance at herd level (based on official diagnosis method), areas with higher prevalence of TB could be better scenarios, allowing to standardize the technique (e.g. establishing percentage or number of animals tested to detect MTC DNA at herd level) and to find out more robust conclusions about the interprtation of this technique.

Reviewer #3: Jordi MARTINEZ-GUIJOSA and collaborators are reporting upon the detection of Mycobacterium tuberculosis complex (MTC) mycobacteria in herds and wild fauna in one region of Spain, where bovine tuberculosis remains of concern.

The topic is of prime interest, yet methods used to address the question are very confusing so that this reviewer had difficulties to draw any firm conclusions from the partial data herein reported.

In details:

1. The tentative integrated approach by the authors (observation of the geography, hygrometry, flora, fauna, etc …) is appropriate and very interesting, but why not sample soil which has been shown to preserve MTC mycobacteria for long, hence a potential source of contamination for the wild and farmed animals? It is a major drawback in the interpretation of reported data; and probably a missed opportunity to better understand the puzzling epidemiology of bovine tuberculosis in that region.

2. The methods used to assess the presence (absence) of MTC in animals are disparate, not all tested animals have been tested with the same methods and in fact, very few animals have been really tested.

3. Among the methods used to assess the presence of MTC in animals, only culture is appropriate, catching living mycobacteria which are presumably the only ones of interest in the natural history of bovine tuberculosis. In fact, only 57 animals have been tested by culture, and only dead animals have been tested. Why not test feces which are a valuable source for MTC DNA and viable mycobacteria ?

4. Methods of interest must be detailed for the reader to interpret data:

a. PCR: Negative controls? Internal DNA extraction control ? calibration curve ? Cts values ?

b. Culture: culture medium ? temperature and atmosphere of incubation ? identification of colonies ?

Minor remarks:

1. Epidemiological diagnosis: this new concept is very unclear for the reviewer. In the opinion of the reviewer, the authors made a (limited) diagnosis of bovine tuberculosis, using three different methods. Interpretation of these limited diagnosis data in terms of epidemiology would require a large sample of tested animals, and integration of the sensitivity / specificity values of each one of the three tests, in calculations.

2. Line 75: Is detection of “environmental” MTC DNA really innovative?

3. Line 134: UCLM ?

7. PLOS authors have the option to publish the peer review history of their article (what does this mean?). If published, this will include your full peer review and any attached files.

Reviewer #1: Yes: José Manuel Benítez-Medina

Reviewer #2: No

Reviewer #3: No

---

## [Author Response · Author response to Decision Letter 1]

29 Mar 2020

Reviewer #1: Thanks for including the suggestions. I believe that paper is now better understood. In any case, as you have clarified, this research is especially a methodological work that must be continued to clarify some issues and prove their true usefulness. Please review references 1 (2015; 45: 160–175.) and 3 (name bacteria in italics).

->We appreciate the review. Changes made.

Reviewer #2: Dear authors,

Many thanks for responding all my comments included in the first round. I maintain that environmental MTC DNA detection is a promising and an interesting tool to be used in epidemiological studies. However, despite of modification included in this new version, data showed in the manuscript do not support suggestions and conclusions yet, and hence, I cannot recommend publication.

In this work, two criteria have been used to analyse TB status of the farms: TB testing results (based on official data between 2001 and 2016), and on-animal detection of environmental MTC DNA at herd level. Detection of MTC DNA was related with three epidemiological factors (probably correlated among them): altitude of the farm, distance to woodland and the use of regional pasture. However, TB risk level does not correlated with any of the risk factors proposed. In addition, no association was detected between TB risk level (based on data from 2001-2016) and environmental MTC DNA detection at herd level.

->Thank you very much for offering us a sincere and critical review. It will help us improve the quality of work.

First, you are correct; there is a correlation between the use of regional pastures and the distance to forest (ANOVA p=0.021), but not with altitude (ANOVA p=0.448). This implies that the different factors that may pose a risk of exposure to mycobacteria are related and, with the data obtained in this study, it is not possible to discern which is the most determining factor.

However, as they are logically related factors, and with a potential risk of exposure to mycobacteria endowed with a biological sense, we cannot rule out any of them in our epidemiological diagnosis, and we believe that all three should be taken into account to take preventive measures to avoid new infections in cattle herds.

This new information is added in lines 245-246 and 258-262.

These results are prudently discussed in few sections of the paper. However, authors maintain in the manuscript some statements like “This study suggests that on-animal environmental DNA sampling can be a proxy for contact (or contact risk) with MTC in livestock, at the herd level”, “Our results allow us to accept the initial hypothesis: the detection of environmental DNA in cattle could contribute to risk factor identification at the cattle herd level and, always in combination with the official TB tests of individual animals, could represent a useful epidemiology tool to investigate MTC maintenance in TB hotspot areas” or “We consequently suggest that MTC DNA could be a good proxy for current infection risk. The low prevalence and incidence in the area and the apparently random detection of positive tests could indicate an environmental origin of the new cases”; which, from my point of view, are not supported by data showed in the manuscript.

->Thank you for the comment. In no case, our statements are intended to be overblown and, therefore, have been modified to clarify this point (lines 36-38, 249-250 and 258-262). However, we do believe that the detection of environmental DNA carried out in this study may be a good indicator of the contact risk of cattle with MTC.

To support these suggestions, it would be necessary an association between detection of MTC DNA and TB positivity at herd levels, but it has not been proven. I know that this fact can be due to the sensitivity problems of the diagnostic test used in low prevalence areas. But this fact could be also due to specificity problems of the MTC DNA detection, mainly in areas closed to woodlands where wild boar with a high percentage (26/31 (84%)) of microbiologically unconfirmed TB-like lesions (probably caused by closely related bacteria) are living.

Once validated, on-animal MTC DNA detection would be an useful tool to be implemented in low-prevalence areas like the studied one. However, in order to assess the association between environmental MTC DNA detection and the risk of TB appearance at herd level (based on official diagnosis method), areas with higher prevalence of TB could be better scenarios, allowing to standardize the technique (e.g. establishing percentage or number of animals tested to detect MTC DNA at herd level) and to find out more robust conclusions about the interprtation of this technique.

->Thank you for the comment. It is true that, a priori, we thought that the detection of environmental DNA could be directly related to the positivity of official tests. However, for biological reasons, the risk of contact with environmental MTC will always be higher than the risk of infection, and we hypothesize that, in areas of low incidence such as the one studied, the detection of environmental DNA from the areas used by livestock would make it easier to establish risk factors than the official tests themselves, which, having such a low and diffuse positivity rate, make it difficult to stablish statistical conclusions.

In addition, we have also planned to carry out new analyzes based on the official tests subsequent to this study, but, because management actions were implemented to try to reduce the prevalence in the area, the epidemiological scenario changed, and therefore the conclusions could be incorrect.

Reviewer #3: Jordi MARTINEZ-GUIJOSA and collaborators are reporting upon the detection of Mycobacterium tuberculosis complex (MTC) mycobacteria in herds and wild fauna in one region of Spain, where bovine tuberculosis remains of concern.

The topic is of prime interest, yet methods used to address the question are very confusing so that this reviewer had difficulties to draw any firm conclusions from the partial data herein reported.

In details:

1. The tentative integrated approach by the authors (observation of the geography, hygrometry, flora, fauna, etc…) is appropriate and very interesting, but why not sample soil which has been shown to preserve MTC mycobacteria for long, hence a potential source of contamination for the wild and farmed animals? It is a major drawback in the interpretation of reported data; and probably a missed opportunity to better understand the puzzling epidemiology of bovine tuberculosis in that region.

->Thank you for the comment. Yes, it is a potential weakness in our study. However, we want to clarify our point of view in this regard.

Firstly, it should be noted that the objective of this work is to test a novel on-cattle environmental DNA detection methodology. Our hypothesis is that by sampling the environmental DNA deposited on the animals, the environmental sampling is maximized by detecting the mycobacteria present in the cattle herds, which would be a proxy for the mycobacteria present in the places where they have been for the last days or weeks, contributing to risk factor identification at the herd level.

In addition, when sampling soil (presumably from a point with a potential risk of transmission of the pathogen, such as water or food), it is very difficult to sample it in an integral way (for example, the entire bottom of a water trough or all the banks of a pond), and nor do we make sure to sample the specific risk points that animals use in their normal behavior (it is extensive mountain livestock where animals are poorly managed when they are grazing freely). Through on-cattle environmental DNA detection, we expect that specificity will be lost regarding which specific places are contributing to the transmission of the pathogen, but it allows us to gain sensitivity in terms that we increase the number of relevant points sampled (potentially all recently visited by the herd) at once.

However, we assume that with soil sampling, we would have obtained additional information that would have allowed us to contrast it with our data, thus allowing a more complete view of the epidemiological scenario of the studied region and allowing us to obtain better conclusions. Therefore, we included this information in the “Discussion” section (lines 269-275) to let the reader know that we consider this weakness in our study.

2. The methods used to assess the presence (absence) of MTC in animals are disparate, not all tested animals have been tested with the same methods and in fact, very few animals have been really tested.

3. Among the methods used to assess the presence of MTC in animals, only culture is appropriate, catching living mycobacteria which are presumably the only ones of interest in the natural history of bovine tuberculosis. In fact, only 57 animals have been tested by culture, and only dead animals have been tested. Why not test feces which are a valuable source for MTC DNA and viable mycobacteria ?

->Thank you for the comments 2 and 3. The same method is always used to detect environmental MTC DNA, specifically, this method is described for the first time in this work and represents a new potential tool to help control and eradicate pathogens shared between wildlife and livestock. This methodology is used to infer the risk of contact with environmental MTC DNA at herd level, so it is not intended to establish the presence (absence) of MTC in each animal, and it is not necessary to sample all animals for this purpose.

Additionally, regarding the methods used in official diagnostic tests at the individual level (each cattle is tested, first by intradermal skin test, in some cases IFN-γ is required, and finally they confirm suspicious tests by culture once slaughtered), we, as scientists, cannot intervene in the official methodologies and protocols, and we can only limit ourselves to use the official data gratefully, which we also consider rigorous and reliable.

Finally, we did not tested feces because our goal was not to diagnose specific animals, but to try to infer the risk of contact of a herd with mycobacteria through dust deposited on the animals belonging to this herd. This dust comes from places that these cattle have visited naturally throughout their usual pasture behavior.

4. Methods of interest must be detailed for the reader to interpret data:

a. PCR: Negative controls? Internal DNA extraction control ? calibration curve ? Cts values ?

->Thank you for the comment. In relation to the controls included in this protocol, positive (M. bovis DNA) and negative (distilled water) controls were included for the PCR step. A sentence is included in the manuscript (lines 166): “Positive (M. bovis strain) and negative PCR controls (distilled water) were also included.”

In relation to the calibration curve, this information is not included in the manuscript since an additional manuscript is ready to be sent. To the reviewer´s knowledge, the PCR was optimized using serial dilutions of a M. bovis DNA. The range of Ct values spanned from an average of 18.83 cycles (S.D. = 0.09) at 6.5 ng/reaction to 39.18 cycles (S.D. = 0.27) at 6.5 fg/reaction. The dynamic range of the reaction spanned from 6.5 ng/reaction to 650 fg/reaction, with a R2 of 0.999 and an overall efficiency of 100%.

During the extraction step, an additional negative sample (sponge with the surfactant liquid) was included.

b. Culture: culture medium? temperature and atmosphere of incubation ? identification of colonies ?

->Thank you for the comment. The references related to the culture protocol have been modified and changed to a recent one where the conditions of decontamination, culture media and confirmation of MTBC by PCR were included. A brief description was also added to the manuscript (lines 110-115): “Briefly, approximately 2g of tissue sample was homogenized and decontaminated with an equal volume of 0.75% (w/v) hexadecyl pyridinium chloride solution in agitation during 30 minutes. After centrifugation, pellets were collected with swabs and cultured in Löwenstein-Jensen with sodium pyruvate and Coletsos media (Difco, Spain) at 37ºC up to 3 months. Culture was considered positive when isolates were identified as MTBC by conventional PCR (Wilton and Cousins, 1992) and /or DVR-spoligotyping ((Kamerbeek et al., 1997)”

https://www.ncbi.nlm.nih.gov/pubmed/1282431

https://www.ncbi.nlm.nih.gov/pmc/articles/PMC229700/

Minor remarks:

1. Epidemiological diagnosis: this new concept is very unclear for the reviewer. In the opinion of the reviewer, the authors made a (limited) diagnosis of bovine tuberculosis, using three different methods. Interpretation of these limited diagnosis data in terms of epidemiology would require a large sample of tested animals, and integration of the sensitivity / specificity values of each one of the three tests, in calculations.

->Thank you. Your comment has shown us that we do not make clear the concept of epidemiological diagnosis (ED). 

First of all, we want to clarify that we are not only referring to the characterization of the infection in the studied region, but rather we are referring to the study and analysis of the distribution, patterns and risk factors of health and disease conditions in defined populations of the studied region (at the epidemiological level). In this sense, we consider that the novel tool used in this work can help to risk factor identification at the herd level. This point has been clarified in lines70-78: “An ED consists in identifying the species involved and their TB status, along with their distribution, abundance and management within the study area. In this specific case, we have generated information on wildlife and livestock numbers and their MTC infection status, carried out interviews with farmers and used participatory Geographic Information System (GIS) [19] to investigate farm-related risk factors for the incidence of TB in cattle. The innovative nature of this study lies in the use of environmental DNA detection as a proxy for exposure risk at the farm (herd) level. Environmental DNA detection is a suitable tool by which to detect contamination with MTC in waterholes and other substrates [20–22]. We hypothesized that the detection of environmental DNA in cattle herds would contribute to risk factor identification at the herd level.”

2. Line 75: Is detection of “environmental” MTC DNA really innovative?

->Thank you for the comment. Actually, the detection of environmental MTC DNA is not innovative, what is innovative is the tool used (the pre-hydrated sponges) and the sampling protocol, based on indirect sampling of the environment through sampling the animals that have used the resources in that environment.

3. Line 134: UCLM ?

-> Thank you. Univesidad de Castilla la Mancha (Castilla la Mancha University)

---

## [Editor Report · Decision Letter 2]

24 Apr 2020

PONE-D-19-20068R2

Environmental DNA points at shared pastures and proximity to woodlands as tuberculosis risk factors for cattle in multi-host settings

PLOS ONE

Dear Mr. Martinez-Guijosa,

Thank you for submitting your manuscript to PLOS ONE. After careful consideration, we feel that it has merit but does not fully meet PLOS ONE’s publication criteria as it currently stands. Therefore, we invite you to submit a revised version of the manuscript that addresses the points raised during the review process.

I am so sorry for the huge delay in sending a decision on your work. The first two referees had a completely differed opinion about the suitability of your work to be accepted for publication. A third referee, however, raised some methodological questions but refused to revise a revised version of your work. To my understanding, you work provided some original and interesting information about the interest of considering environmental DNA in TB epidemiological models. However, your sample size is moderate as well as the contribution of the environmental DNA in your epidemiological model. In consequence, I strongly recommend you to get in touch with a native speaker to help you to put your results into context. I am sure that a good English editing will enhance the impact or your findings. On the other hand, I think that “Environmental DNA a promising factor for tuberculosis risk assessment in multi-host settings” is a more appropriate title for your work. Enclosed you will find some additional comments embedded in the pdf.

We would appreciate receiving your revised manuscript by Jun 08 2020 11:59PM. To enhance the reproducibility of your results, we recommend that if applicable you deposit your laboratory protocols in protocols.io, where a protocol can be assigned its own identifier (DOI) such that it can be cited independently in the future. For instructions see: http://journals.plos.org/plosone/s/submission-guidelines#loc-laboratory-protocols

We look forward to receiving your revised manuscript.

Kind regards,

Emmanuel Serrano, PhD

Academic Editor

PLOS ONE

---

## [Author Response · Author response to Decision Letter 2]

13 May 2020

Thank you for submitting your manuscript to PLOS ONE. After careful consideration, we feel that it has merit but does not fully meet PLOS ONE’s publication criteria as it currently stands. Therefore, we invite you to submit a revised version of the manuscript that addresses the points raised during the review process.

I am so sorry for the huge delay in sending a decision on your work. The first two referees had a completely differed opinion about the suitability of your work to be accepted for publication. A third referee, however, raised some methodological questions but refused to revise a revised version of your work. To my understanding, you work provided some original and interesting information about the interest of considering environmental DNA in TB epidemiological models. However, your sample size is moderate as well as the contribution of the environmental DNA in your epidemiological model.

-> Thank you for your consideration. We expect that this new version of the manuscript fulfill all expectations.

In consequence, I strongly recommend you to get in touch with a native speaker to help you to put your results into context. I am sure that a good English editing will enhance the impact or your findings.

-> OK, done.

On the other hand, I think that “Environmental DNA a promising factor for tuberculosis risk assessment in multi-host settings” is a more appropriate title for your work.

-> Thank you for the recommendation. The title has changed to “Environmental DNA: a promising factor for tuberculosis risk assessment in multi-host settings”.

Enclosed you will find some additional comments embedded in the pdf.

-> Thank you. All comments have been considered and included into the new version of the manuscript.

---

## [Editor Report · Decision Letter 3]

14 May 2020

Environmental DNA: a promising factor for tuberculosis risk assessment in multi-host settings

PONE-D-19-20068R3

Dear Dr. Martínez-Guijosa,

We are pleased to inform you that your manuscript has been judged scientifically suitable for publication and will be formally accepted for publication once it complies with all outstanding technical requirements.

With kind regards,

Emmanuel Serrano, PhD

Academic Editor

PLOS ONE

Additional Editor Comments (optional):

My congratulations!

Emmanuel
---

## [Editor Report · Acceptance letter]

19 May 2020

PONE-D-19-20068R3 

Environmental DNA: a promising factor for tuberculosis risk assessment in multi-host settings 

Dear Dr. Martínez-Guijosa:

I am pleased to inform you that your manuscript has been deemed suitable for publication in PLOS ONE. Congratulations! Your manuscript is now with our production department. 

With kind regards,

on behalf of

Dr. Emmanuel Serrano 

Academic Editor

PLOS ONE